# A Review of Recent Developments in the Molecular Mechanisms of Bone Healing

**DOI:** 10.3390/ijms22020767

**Published:** 2021-01-14

**Authors:** Emerito Carlos Rodríguez-Merchán

**Affiliations:** Osteoarticular Surgery Research, Hospital La Paz Institute for Health Research–IdiPAZ, La Paz University Hospital–Autonomous University of Madrid, 28046 Madrid, Spain; ecrmerchan@hotmail.com

**Keywords:** bone, fracture, healing, bone tissue engineering, osteoblasts, osteoclasts, periosteum, osteogenesis, angiogenesis

## Abstract

Between 5 and 10 percent of fractures do not heal, a condition known as nonunion. In clinical practice, stable fracture fixation associated with autologous iliac crest bone graft placement is the gold standard for treatment. However, some recalcitrant nonunions do not resolve satisfactorily with this technique. For these cases, biological alternatives are sought based on the molecular mechanisms of bone healing, whose most recent findings are reviewed in this article. The pro-osteogenic efficacy of morin (a pale yellow crystalline flavonoid pigment found in old fustic and osage orange trees) has recently been reported, and the combined use of bone morphogenetic protein-9 (BMP9) and leptin might improve fracture healing. Inhibition with methyl-piperidino-pyrazole of estrogen receptor alpha signaling delays bone regeneration. Smoking causes a chondrogenic disorder, aberrant activity of the skeleton’s stem and progenitor cells, and an intense initial inflammatory response. Smoking cessation 4 weeks before surgery is therefore highly recommended. The delay in fracture consolidation in diabetic animals is related to BMP6 deficiency (35 kDa). The combination of bioceramics and expanded autologous human mesenchymal stem cells from bone marrow is a new and encouraging alternative for treating recalcitrant nonunions.

## 1. Introduction

It is estimated that between 5% and 10% of bone fractures do not heal properly [1]. The categories of bone fracture are the following: Closed or open fractures; complete fractures; displaced fractures; partial fractures; stress fractures. Some extra terms must also be added to describe partial, complete, open, and closed fractures. These terms include avulsion; comminuted; compression; impacted; oblique; spiral; transverse. Most often, bone fractures happen because the bone runs into a stronger force. Repetitive forces, such as running, can also fracture a bone (stress fractures). Another reason for fractures is osteoporosis, which weakens bones as you age.

Internal fixation for nonunions should provide sufficient stability for fracture healing without excessive rigidity. The choice of internal fixation depends on the type of nonunion, the condition of the soft tissues and bone, the size and position of the bone fragments, and the size of the bony defect [2]. Several biological enhancement methods have been published so far for managing nonunions (Table 1) [3,4,5].

There are, however, no current pharmacological treatments to enable effective bone consolidation. A better understanding of the molecular mechanisms underlying bone healing is therefore essential for developing new treatments to accelerate the process [1].

A biological or mechanical deficiency, a lack of information regarding the host’s comorbidities, and a lack of vascularization can all lead to nonunion. The presence of osteoinductive mediators, osteogenic cells, and an osteoconductive matrix (scaffolding) is paramount for proper unions. An optimal mechanical environment, appropriate vascularization, and treatment of any pre-existing comorbidity are also required for proper unions [4].

In this article the following topics will be analyzed: The role of a specific population (Prx1+ Cells) and its expression marker (Prx1) in fracture repair; integration between the cytoskeleton and the main molecular pathways in relation to the mechanotransmission mechanisms in osteocytes; factors that induce bone formation (type H vessels, endogenous-exogenous combined bionic periosteum, after BMP2 stimulation preconditioned cells display increased in-vivo bone formation ability, in vivo sequestration of innate small molecules, phagocytic role of macrophages, morin); inducing bone formation with mesenchymal stem cells (leptin, two protein networks, Trb3, Interleukin-1β); factors that delay bone healing (inhibition of estrogen receptor alpha signaling, smoking); studies on diabetes and osteoporosis; an improved method for assessing cell and molecular signals in the reparative callus during fracture healing; a novel experimental model to study fracture healing of the proximal femur; co-culture systems of osteoblasts and osteoclasts (Table 2).

A recent article showed that bone healing was efficaciously achieved in 28 patients with nonunions surgically treated with a combination of autologous expanded mesenchymal stromal cells from human bone marrow (hBM-MSCs) (between 100 × 10^6^ and 200 × 10^6^ expanded hBM-MSCs) combined with a bioceramic [MBCP + TM, a 100% synthetic (Biomatlante, Vigneux, France) bone substitute composed of 20% hydroxyapatite and 80% beta-tricalcium phosphate, in 1–2 mm granules]. The cell-biomaterial association was performed in the surgical room prior to implantation [5].

The purpose of this article is to review the most recent advances in the molecular mechanisms of bone consolidation. We conducted a bibliographic search on the 29th of December 2020 in PubMed using “molecular mechanism bone healing” as the keywords, resulting in 753 articles, 82 of which we ultimately analyzed, given that they were directly related to the molecular mechanisms of bone healing.

## 2. The Role of a Specific Population (Prx1 + Cells) and Its Expression Marker (Prx1) in Fracture Repair

The bone’s capacity to heal after a fracture depends on adult regenerative cells; however, information on the identification and functional role of fracture-induced adult regenerative cells is still lacking [6,7,8,9,10,11,12,13,14]. During the skeleton’s genesis, the paired-related homeobox (Prx1) is expressed. Esposito et al. recently showed that a fracture reinstitutes Prx1 expression and that Prx1-expressing cells are essential for inducing bone repair. The authors also showed that fractures cause an early increase in bone morphogenetic protein (BMP)-2 expression, followed by a decrease in C-X-C motif chemokine ligand 12 (CXCL12) levels (a soluble ligand secreted by bone marrow stromal cells that stimulate proliferation and growth of B-cell progenitors), which in turn downregulates Prx1, allowing the cells to engage in osteochondrogenesis. In vivo and in vitro treatment with the chemokine antagonist CXCR4 AMD3100 (Plerixafor) enables the restoration of Prx1 expression by modulating the BMP-2-CXCL12 axis. Esposito et al., therefore, defined the role of a specific population (Prx1+ cells) and its expression marker (Prx1) in fracture repair, information that could be useful in the search for new treatments for nonunion fractures [15].

## 3. Integration between the Cytoskeleton and the Main Molecular Pathways in Relation to the Mechanotransmission Mechanisms in Osteocytes

Bone regeneration happens by two mains ossification processes, endochondral ossification in which the skeletal element first develops as a cartilaginous template that is subsequently replaced by bone, and intramembranous ossification in which mesenchymal cells directly differentiate into bone-forming osteoblasts. The ossification process does not need pre-existing cartilage. Both procedures need the replenishment of the osteogenic or chondrogenic progenitor cells that participate in bone or cartilage formation during normal development and under pathologic conditions, such as fracture healing. In general, osteogenic progenitors distribute in various bone compartments along the bone’s outer surface within the periosteum and the inner surface of bone within the endosteum.

To better understand how fractures heal, it is important to know the cellularity and metabolism of bone tissue, especially its cytoskeletal architecture, as well as its transformations due to external mechanical stimuli. The specific physical and chemical characteristics of the extracellular matrix (ECM) enable these stimuli to be transmitted from the outside of the cell to the plasma membrane. There has been growing interest in osteocytes in relation to bone homeostasis due to their role in coordinating other cell populations. The cytoskeleton of osteocytes consists of a complex network of actin and microtubules combined with crosslinker proteins such as vinculin and fimbrin, which connect and transmit external stimuli to the cytoplasm via the ECM. As a result of mechanical stimuli, important signaling pathways are activated in the cytoplasm, such as those dependent on Cx43, MAPK/ERK, Wnt, YAP/TAZ, and Rho-ROCK. This activation leads to cytoskeletal changes in the osteocytes and remodeling of the ECM, circumstances that alter the bone tissue. The latest advances in intracellular signaling pathways, the osteocyte mechanotransmission system, and bone tissue engineering suggest promising experimental strategies, some of which could be employed in clinical trials [16].

## 4. Factors That Induce Bone Formation

Factors that can induce bone formation are the following: Patient-related factors (age, osteoporosis, smoking, drugs); bone grafting, biochemical stimulation (growth factors, bone morphogenetic proteins, transforming growth factor-beta); cell-mediated bone healing (osteogenic stem cells); gene therapy (transfecting bone stem cells with genes encoding for growth factors that stimulate bone healing); physical factors (electrical stimulation). In this review, we only discuss a few ones described in the recent development.

### 4.1. Type H Vessels Induce Bone Formation

Angiogenesis is closely connected to osteogenesis throughout skeletal development and in the course of bone modeling and remodeling. Blood vessels do not solely provide bone tissues with the required nutrients, oxygen, growth factors, and hormones, they have been found to play a paramount role in the regulation of bone formation [17,18,19,20,21,22,23,24,25,26,27].

Type H vessels, so named for their high expression of endomucin and CD31, have recently been identified as able to induce bone formation. Factors such as platelet-derived growth factor type BB, slit guidance ligand 3, hypoxia-inducible factor 1-alpha, Notch, and vascular endothelial growth factor (VEGF) are involved in the combination of angiogenesis and osteogenesis [28].

### 4.2. Endogenous-Exogenous Combined Bionic Periosteum Is Effective and Adaptable in Activating Periosteal and Bone Regeneration

The periosteum plays a key role in the development and healing of bone injuries. Wu et al. demonstrated that endogenous-exogenous combined bionic periosteum is effective and versatile in triggering periosteal and bone regeneration, a promising finding for cases of bone nonunion [29,30]. Wu et al. employed a hierarchical micro/nanofibrous bionic periosteum with sustained VEGF release and an exogenous vascularized fibrous layer of periosteum to induce an endogenous cambium layer in vivo that completely regenerated the periosteum and bone tissue, using collagen self-assembly and microsol-electrospinning technologies. The authors demonstrated that the VEGF-encapsulated hyaluronan-poly(L-lactic acid) core-shell structure was durably steadily released during angiogenesis in the fibrous layer and the bone defect. Meanwhile, the self-assembly of collagen with electrospun fibers contributed to a hierarchical micro/nanostructure that largely mimicked the microenvironment of the ECM, allowing for cell adhesion, proliferation, and differentiation and leading to the formation of a cambium layer that mimicked in-situ ossification in the same manner as occurs in intramembranous ossification [31].

### 4.3. After BMP2 Stimulation, Preconditioned Cells Display Increased In-Vivo Bone Formation Ability

The success of cell-based constructs developed for regenerative medicine has been hindered considerably due to a nondependable in vivo biological potency of the implant. This has been associated with a lack of in-depth characterization of the in vitro product and its relation to in vivo potency [32,33,34,35,36,37,38].

Bolander et al. investigated the cellular and molecular changes associated with a serum-free preconditioning-induced change in the cell phenotype of human periosteum-derived cells. After BMP2 stimulation, the preconditioned cells showed an increased capacity for in vivo bone formation, which was associated with adapted cell metabolism together with elevated BMPR2 (BMP receptor type 2) expression. Single-cell RNA sequencing confirmed the activation of the pathways and transcriptional regulators involved in bone development and fracture healing, leading to an increase in the specified skeletal progenitor cell populations. These findings show the importance of appropriate in vitro conditions to achieve good results in vivo. In addition, BMPR2 appears to be a promising biomarker for enriching skeletal progenitor cells to achieve in vivo bone regeneration [39].

### 4.4. In Vivo Sequestration of Innate Small Molecules Promotes Bone Healing 

Methods that facilitate innate repair mechanisms hold great potential for tissue repair [40]. Zeng et al. described the biomaterial-assisted sequestration of small molecules to locate proregenerative signaling at the fracture site, employing a synthetic biomaterial containing boronated molecules designed to sequester adenosine (a small molecule present in the human body). Biomaterial-assisted adenosine sequestration takes advantage of the transient increase in extracellular adenosine after a fracture to prolong local adenosine signaling. The implantation of the biomaterial patch after a fracture establishes an in situ adenosine reserve, which accelerates consolidation by promoting osteoblastogenesis and angiogenesis. The patch’s adenosine content decreases to physiological levels as the bone tissue regenerates. In addition to sequestering endogenous adenosine, the biomaterial is capable of carrying exogenous adenosine to the fracture site. The patch, therefore, appears to be a good therapeutic alternative for bone tissue repair [41].

### 4.5. The Phagocytic Role of Macrophages in Various Periods of Bone Consolidation

The coordinated interaction between osteogenesis and the osteoimmune microenvironment is essential for the proper consolidation of a bone fracture, with macrophages playing an essential regulatory role in all stages of bone repair. Depending on the signals they detect, macrophages can mediate the host’s immune response against external signals from molecular stimuli and implanted scaffolds, thereby exercising their regenerative capacity to varying degrees [35,42,43,44,45,46,47,48].

In one study, Niu et al. analyzed three aspects of the immunomodulatory functions of macrophages during bone regeneration: as sweepers, mediators, and instructors. The authors described the phagocytic role of macrophages during specific periods of bone consolidation (“sweepers”) and the variety of paracrine cytokines released by macrophages, either by mediating cell mobilization, vascularization, and matrix remodeling (“mediators”) or by directly promoting osteogenic differentiation of bone progenitors and bone repair (“instructors”). The authors aimed to exploit the power of endogenous macrophages to enhance the performance of engineered bone tissue [49].

### 4.6. Morin Has Pro-Osteogenic Capacity

Wan et al. observed that the natural compound morin (a pale yellow crystalline flavonoid pigment [C_15_H_10_O_7_] found in old fustic and osage orange trees) has pro-osteogenic capacity. In one study [50], the authors employed in vivo and in vitro models to investigate the mechanisms of morin’s biological osteoblastic activity at the molecular level. By administering morin to skull-defected mice at a purity of ≥95% (as determined by high-performance liquid chromatography) and a dosage of 100 mg/kg/day, the authors evaluated the efficacy of morin in pro-osteogenesis by monitoring the changes in bone histomorphometry scores, the development of immature osteoblasts from MSCs, and the increased expression of pro-osteogenic cytokines. The authors employed quantitative polymerase chain reactions, western blot analysis, and immunofluorescence to investigate the signaling pathways. The authors showed that morin had an important pro-osteogenic effect in vivo that might facilitate osteoblast development and the production of osteoblast-related marker genes and in vitro protein markers for osteoblasts. In terms of molecular biology, morin contributes to osteoblast development and Wnt pathway stimulation through the activation and translocation of β-catenin nuclei. In short, morin can be a good bone substitute that can provide benefits for regenerating bone defects [50].

### 4.7. Studies on Inducing Bone Formation with Mesenchymal Stem Cells

#### 4.7.1. Leptin Might Potentiate BMP9-Induced Osteogenesis by Cross-Regulating BMP9 Signaling through the JAK/STAT Signaling Pathway in Mesenchymal Stem Cells

Discovered around 25 years ago as an adipocyte-derived hormone created in direct proportion to the quantity of body fat, leptin has pleiotropic functions and regulates food intake, energy metabolism, the reproductive system, inflammation, immunity, and bone mass and mineral density [51,52,53,54,55,56].

Zhang et al. investigated the possible effects of leptin signaling on the BMP9-induced osteogenic differentiation of MSCs [57]. Leptin is a hormone derived from adipocytes in direct proportion to the amount of body fat and exerts pleiotropic functions such as the regulation of energy metabolism, bone mass, and mineral density. The authors found that exogenous leptin potentiates the BMP9-induced osteogenic differentiation of MSCs in vitro and in vivo while inhibiting BMP9-induced adipogenic differentiation. BMP9 was shown to induce leptin expression and increase leptin receptor levels in MSCs, while exogenous leptin increased BMP9 expression in less differentiated MSCs. The authors demonstrated that a blockade of JAK signaling effectively attenuated BMP9-induced leptin-potentiated osteogenic differentiation. These findings appear to indicate that leptin potentiates BMP9-induced osteogenesis by cross-regulating BMP9 signaling through the JAK/STAT signaling pathway in MSCs. It, therefore, seems logical that the combined use of BMP9 and leptin might improve bone regeneration and fracture healing [57].

#### 4.7.2. Two Protein Networks Are Potentially Implicated in Osteoinduction

Studies of adsorption of serum proteins on biomaterials and protein production by cells in contact with biomaterials can give valuable data about the relationship between material properties and function [58], especially taking into account that protein adsorption is the first interaction between a material and a biological system following implantation. Furthermore, it is the production of proteins by adherent cells that eventually determines cell fate and consequently the efficacy of regeneration [58,59].

Othman et al. identified two protein networks potentially involved in the osteoinduction process, one consisting of collagen fragments and collagen-related enzymes and another consisting of endopeptidase inhibitors and regulatory proteins. The authors’ results show that protein profiling is a useful tool for understanding the interactions between a biomaterial and a biological system, which could help design and develop better biomaterials for use in bone regenerative therapies [60].

#### 4.7.3. Trb3 Controls Mesenchymal Stem Cell Lineage Fate and Ameliorates Bone Regeneration by Scaffold-Mediated Local Gene Delivery

Several publications have suggested that Trb3 is a promising molecular target to regulate adipo-osteogenic differentiation of MSCs [61,62,63,64,65,66,67,68,69]. Fan et al. investigated the reciprocal role of Trb3 in regulating the osteogenic and adipogenic differentiation of MSCs in the context of bone formation and examined the mechanisms by which Trb3 controls the adipogenic-osteogenic balance. Trb3 is a member of the tribbles family of pseudokinases. Trb3 promoted the osteoblastic commitment of MSCs at the expense of adipocyte differentiation. Mechanically, Trb3 regulated the choice of MSC fate through BMP/Smad and Wnt/β-catenin signals. The authors observed that the local in vivo administration of Trb3 via a gelatin-conjugated caffeic acid-coated apatite/poly(lactide-co-glycolide) scaffold stimulated robust bone regeneration and inhibited fat-filled cyst formation in rodent non-healing mandibular defect models. These findings show that homolog tribble (Trb3)-based therapeutic strategies favor osteoblastogenesis over adipogenesis, which can be used to improve skeletal regeneration and the future treatment of diseases with bone loss [70].

#### 4.7.4. Interleukin-1β Promotes Osteogenic Differentiation and Function of Mouse Bone Marrow Mesenchymal Stem Cells via the BMP/Smad Signaling Pathway

Wang et al. suggested that, within a certain concentration range, interleukin (IL)-1β promotes the differentiation and osteogenic function of mouse bone marrow MSCs (MBMMSCs) through the bone morphogenetic protein/Smad (BMP/Smad) signaling pathway [71]. MBMMSCs proliferation in the presence of IL-1β was analyzed utilizing a Cell-Counting Kit-8 assay, and the effect of IL-1β on MBMMSC apoptosis was studied via flow cytometry. Alkaline phosphatase assay, Alizarin Red staining, and quantitative assays were carried out to assess the osteogenic differentiation of MBMMSCs. The expression levels of osteogenic differentiation markers were detected utilizing reverse transcription-quantitative PCR (RT-qPCR). It was shown that within a concentration range of 0.01-1 ng/mL, IL-1β promoted osteogenic differentiation of MBMMSCs and did not induce apoptosis. Moreover, RT-qPCR results indicated that IL-1β augmented osteogenic gene expression within this concentration range. Moreover, Western blotting results identified that the bone morphogenetic protein/Smad (BMP/Smad) signaling pathway was significantly activated by IL-1β under osteogenic conditions.

## 5. Factors that Delay Bone Healing

Factors that delay bone healing can be divided into local and systemic. The most important local factors are the following: inadequate bone reduction, unstable bone fixation, bone infection, and radiation. The most important systemic factors are the following: patient age (bone healing is faster in children than in adults), nutrition status (sufficient amount of nutrients and vitamins A, B, C, and D are essential for the healing of broken bones). Smoking has a negative effect on bone healing. Steroids also can slow down the healing process. Systemic diseases such as hyperthyroidism and renal insufficiency delay fracture healing. Genetic diseases such as Marfan syndrome, Ehler–Danlos syndrome, osteogenesis imperfecta are among the factors affecting bone healing.

### 5.1. Inhibition of Estrogen Receptor Alpha Signalling Delays Bone Regeneration

Wu et al. evaluated the role of the estrogen receptor alpha (ERα) axis in bone consolidation and its possible mechanisms of action, demonstrating that inhibition of the ERα signaling delays bone regeneration. Female Institute of Cancer Research (ICR) mice were bred with a metaphyseal bone defect in the left femur and were administered methyl piperidino pyrazole (MPP), an ERα inhibitor, and bone consolidation was evaluated by microcomputer tomography. ERα placement with alkaline phosphatase (ALP) and ERα translocation into the mitochondria were determined, and the levels of ERα, ERβ, PECAM-1, VEGF, and β-actin were measured. The expression of chromosomal Runx2, ALP, and osteocalcin mRNAs and mitochondrial cytochrome c oxidase (COX) I and COXII mRNAs was quantified. Angiogenesis was measured by immunohistochemistry. After surgery, the bone mass was increased in the bone-defect area in a time-dependent manner. Simultaneously, the ERα levels increased, correlating positively with bone consolidation. The administration of MPP decreased ERα levels and bone consolidation. Regarding the mechanism of action in bone consolidation, osteogenesis was improved; however, MPP attenuated osteoblast maturation. In parallel, the expression levels of osteogenesis-related ALP, Runx2, and osteocalcin mRNAs were increased in the injured zone. Treatment with MPP produced significant inhibition of the expression of ALP, Runx2, and osteocalcin genes, decreased translocation from ERα to the mitochondria, and expression of COX-1 and COX-2 genes related to mitochondrial energy production, and decreased levels of PECAM-1 and VEGF in the area of the experimentally created bone defects. The study demonstrated the role of the ERα axis in bone consolidation through the stimulation of energy production, osteoblast maturation, and angiogenesis [72].

### 5.2. Smoking Alters Inflammation and Skeletal Stem and Progenitor Cell Activity during Fracture Healing

Smoking causes delayed union and/or nonunion of bone fractures. Unfortunately, orthopedic surgeons rarely delay surgery in patients who smoke nor do they suggest methods for patients to quit smoking. It is important to recommend smoking cessation methods such as transdermal patches, chewing gum, lozenges, inhalers, sprays, bupropion, and varenicline during the perioperative period. Smoking cessation in the perioperative period appears to be effective in reducing delayed union and nonunion rates of bone fractures, even if performed up to 4 weeks prior to the surgery [73].

Hao et al. published a study in which they exposed three murine strains (C57BL/6J, 129 × 1/SvJ, and BALB/cJ) to cigarette smoke for 3 months before performing a midshaft transverse femoral osteotomy. Using radiography, microcomputed tomography, and biomechanical tests, the authors evaluated fracture healing 4 weeks after the osteotomy. The radiographic study showed a significant decrease in the fracture healing capacity of 129 × 1/SvJ smoke-exposed mice. The microcomputed tomography results showed a delay in the remodeling of the fracture calluses in all three strains after exposure to cigarette smoke. The biomechanical tests showed a more significant deterioration of functional properties in the 129 × 1/SvJ mice than in the C57BL/6J and BALB/cJ mice after exposure to cigarette smoke. In other words, the 129 × 1/SvJ strain was the most suitable for simulating the smoke-induced deterioration of fracture healing. In the 129 × 1/SvJ mice, the authors investigated the molecular and cellular disorders of fracture healing caused by cigarette smoke using histology, flow cytometry, and multiplex cytokine/chemokine analysis. The histological analysis showed abnormal chondrogenesis due to cigarette smoke exposure. In addition, significant populations of repair cells, including skeletal stem cells and their subsequent progenitors, showed a decrease in post-injury expansion as a result of cigarette smoke exposure. Furthermore, the authors observed a significant increase in pro-inflammatory mediators and immune cell recruitment in fracture hematomas in the mice exposed to smoke. These results show the important cellular and molecular disorders that occur during fracture healing due to smoking, such as abnormal chondrogenesis, aberrant activity of skeletal stem and progenitor cells, and an intense initial inflammatory response [74]. Table 3 shows the main factors that induce and delay bone healing according to recent publications on the molecular mechanisms of bone healing.

## 6. Studies on Diabetes and Osteoporosis

### 6.1. Transcriptomic and Proteomic Approaches to Bone Regeneration Research in Relation to Type 1 Diabetes and Osteoporosis

Transcriptomic and proteomic approaches have shown great potential in terms of bone regeneration because they offer new knowledge on the molecular physiological/pathological mechanisms that regulate bone consolidation [75,76]. Transcriptomic and proteomic approaches in bone regeneration research, particularly in relation to type 1 diabetes and osteoporosis, appear to be important in current practice [77].

### 6.2. Delays in Fracture Healing in Diabetic Animals Are Related to BMP6 Deficiency (35 kDa)

Using a diabetic rodent model, Guo et al. investigated the relationship between BMP6 and BMP9 and the effect of bone consolidation in diabetes. The authors analyzed the difference in size and calcification of the calluses of fractures and the mechanical resistance and expression of BMP6 and BMP9 in the calluses. The authors evaluated the consolidation of femoral fractures by quantifying callus size and calcification using X-rays, histological and histochemical imaging, the load capacity of the fractured bone, and the amount of BMP6 in the calluses and bones using western blotting. At the end of the second and fourth weeks after the fracture, the authors observed a significant increase in BMP6 in the calluses and fractured bones in both the nondiabetic and diabetic animals. However, the authors detected significantly lower levels of BMP6 (35 kDa) with smaller calcified callus sizes and low load-bearing capacity of bones with already fused fractures in the diabetic animals compared with the nondiabetic animals. A deterioration in BMP6 (35 kDa) maturation from its precursors could be the cause of the decrease in BMP6 in diabetic animals. Therefore, it seems that the delay in fracture healing in the diabetic animals is related to a BMP6 deficiency (35 kDa), which could be due to a defect in BMP6 maturation from its precursors to its functional format [78].

### 6.3. Osteoporotic Fractures in Older Adults Are Associated with a Skeletal Stem Cell Defect

Ambrosi et al. studied whether highly purified bona fide human skeletal stem cells (hSSCs) isolated from geriatric fractures showed intrinsic functional defects that would prevent them from healing. Using flow cytometry, the authors analyzed and isolated hSSCs from the calluses of 61 fractures from 5 different skeletal areas of patients aged 13 to 94 years to conduct functional and molecular studies. The authors observed that fracture-activated amplification of hSSC populations was comparable at all ages. However, the functional analysis of the isolated stem cells revealed that advanced age was significantly related to a reduction in osteochondrogenic potential but not to a decrease in clonogenicity in vitro. HSSCs derived from women displayed an exacerbated functional decline with age relative to those of older men. Transcriptomic comparisons showed downregulation of skeletogenic pathways such as WNT and upregulation of senescence-related pathways in younger versus older hSSCs. The loss of sirtuin1 expression played an important role in hSSC dysfunction, although reactivation by trans-resveratrol or a small molecular compound restored the differentiation potential in vitro. These findings demonstrate the age-related defects in purified hSSCs in geriatric fractures and could serve as a basis for further research into the functional mechanism and reversibility of skeletal stem cell aging in humans [79].

## 7. An Improved Method for Assessing Cell and Molecular Signals in the Reparative Callus during Fracture Healing

Valiya Kambrath et al. published an improved method for the early-stage processing of fracture calluses and for the immunofluorescence labeling of sections to visualize the timing and spatial patterns of cellular and molecular events that regulate the healing of bone fractures. This method does not require prolonged decalcification, so the response time from sample collection to microscopy is short. Furthermore, the method preserves the structural integrity of the fragile callus, because it does not involve deparaffinization or harsh antigen retrieval methods. This method could be adapted for high-throughput screening of drugs that promote the healing of bone fractures [1].

## 8. A Novel Experimental Model to Study Fracture Healing of the Proximal Femur

Haffner-Luntzer et al. published a new experimental mouse model to study the consolidation of metaphyseal fractures of the proximal femur. Their technique consisted of inserting a 24G needle into the femur in a closed fashion and subsequently performing an open Gigli 0.4-mm saw osteotomy of the proximal femur. The authors analyzed the fractured femurs by microcomputed tomography and histology 14 and 21 days after the surgery. All of the study animals showed normal limb loading and a physiological gait pattern in the first three days after the fracture. Robust endochondral ossification was observed during the fracture healing process, with high expression of late chondrocytes and early osteogenic markers on day 14. On day 21, all fractures had a bony bridging score of 3 or more, indicating successful healing. Callus volume decreased significantly from day 14 to day 21, while a high number of osteoclasts appeared in the fracture callus until day 21, indicating that callus remodeling had already begun on day 21. Thus, the authors developed a novel experimental model in mice that enables the consolidation of endochondral fractures of the proximal femur to be studied. This model could be useful for future research using transgenic animals to better understand the molecular mechanisms of consolidation in metaphyseal osteoporotic fractures [80].

## 9. Co-Culture Systems of Osteoblasts and Osteoclasts: Simulating In-Vitro Bone Remodeling in Regenerative Approaches

Co-culture systems allow us to explore interactions between cells. In addition, the replication of naturally occurring cells in multicellular tissues can help design reliable bone engineering models. However, there is still no optimal form of 3D co-culture of human bone cells that accurately reproduces the bone microenvironment. In vitro co-culture systems using human cells might, in the future, become a valid alternative to animal studies. In fact, we can create a mechanobiological environment using dynamic 3D co-culture models. In addition, the basic regenerative mechanisms of bone can be identified using new imaging modalities. Currently, in silico models provide data for adjusting the dynamic parameters applied to the culture systems, thereby better imitating the response of native tissues to scaffolds. Current in vivo models are taking into account various factors related to the healing site, such as the presence of macrophages, the angiogenic process, and the interaction between regenerative cells and immune counterparts [81].

## 10. Conclusions

Bone tissue engineering appears promising, although its success often depends on a “smart” scaffold to host and guide bone formation through the precursors of bone cells. Bone homeostasis basically depends on osteoblasts and osteoclasts in a continuous cycle of bone resorption and formation. Studies on the periosteum are essential, given that it plays a crucial role in bone development and the process of fracture healing. The role of macrophages as central regulators during all phases of bone repair also needs further investigation. Research into the relationship between osteogenesis and angiogenesis is essential because they are intimately linked during bone growth and regeneration in bone modeling and during bone homeostasis in bone remodeling. The role of leptin as an enhancer of osteogenesis induced by BMP9 should be further explored through the cross-regulating BMP9 signaling by the JAK/STAT signaling pathway in MSCs. Lastly, the role of morin, which could be beneficial in the regeneration of bone defects, should also be further studied. A combination of autologous MSCs expanded from bone marrow and synthetic scaffolding (commercially existing biphasic calcium phosphate bioceramic granules) appear to be good options. A combination of bioceramics and expanded autologous hBM-MSCs is an encouraging new treatment option for recalcitrant nonunions in the clinical setting.

## Figures and Tables

**Table 1 ijms-22-00767-t001:** Main methods of biophysical enhancement in treating bone nonunions.

Bone autograft
Bone allograft (demineralized, cancellous, cortical)
Demineralized bone matrix
Reamer-irrigator-aspirator system
Bone substitutes formed by collagen scaffolds, hydroxyapatite, and tricalcium phosphate
Pulsed electromagnetic fields
Low-intensity pulsed ultrasound
Extracorporeal shock waves
Percutaneous injection of autogenous bone marrow
Platelet-rich plasma
Bone morphogenetic proteins
Stem cells: bone marrow aspirate
Biphasic calcium phosphate bioceramic granules combined during surgery with autologous mesenchymal stem cells expanded from bone marrow

**Table 2 ijms-22-00767-t002:** Brief description of the different elements on fracture healing analyzed in this article.

**The Role of a Specific Population (Prx1 + Cells) and Its Expression Marker (Prx1)**
**Integration between the cytoskeleton and the main molecular pathways in relation to the mechanotransmission mechanisms in osteocytes**
**Factors that induce bone formation** (1) Type H vessels (2) Endogenous-exogenous combined bionic periosteum (3) Preconditioned cells (after BMP2 stimulation) (4) In vivo sequestration of innate small molecules (5) Macrophages (phagocytic role) (6) Morin (7) Mesenchymal stem cells (leptin; two protein networks; Trb3; interleukin-1β)
**Factors that delay bone healing** (1) Inhibition of estrogen receptor alpha signaling (2) Smoking
**Studies on diabetes and osteoporosis** (1) Transcriptomic and proteomic approaches (2) BMP6 deficiency (35 kDa); (3) Skeletal stem cell defect
**An improved method for assessing cell and molecular signals in the reparative callus**
**A novel experimental model to study fracture healing of the proximal femur**
**Co-culture systems of osteoblasts and osteoclasts**

**Table 3 ijms-22-00767-t003:** Factors that induce and delay bone healing according to recent publications on the molecular mechanisms of bone healing.

**Factors That Induce Bone Healing**
Type H vessels
Endogenous-exogenous combined bionic periosteum
Bone morphogenetic protein receptor type 2
Patch of synthetic biomaterial containing boronate molecules
Paracrine cytokines released by macrophages
Morin (a pale yellow crystalline flavonoid pigment [C_15_H_10_O_7_] found in old fustic and osage orange trees)
The combined use of bone morphogenetic protein-9 and leptin
Interleukin-1β
**Factors That Delay Bone Healing**
Inhibition of the estrogen receptor alpha signaling
Tobacco smoking
Deficiency of bone morphogenetic protein-6 (35 kDa)]

## Data Availability

Not applicable.

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
