# Peer review of "A Review of Recent Developments in the Molecular Mechanisms of Bone Healing"

_ijms, 2021, doi:10.3390/ijms22020767_

Round 1

Reviewer 1 Report

This review manuscript covers recent development in the molecular mechanisms of bone healing. The authors discussed a few key factor including proteins, genes, influences, and environmental factors related to bone fracture repair. The review was not organized very well and it may need an extensive revision. 

  1. The authors should add a short paragraph summarizing the categories of bone fracture and possible reasons causing bone to break in the introduction.
  2. The authors may add a short paragraph in the introduction to summarize proteins, genes and other factors that they discussed in the later sections, as they did in the conclusion. In this way, it will give readers a comprehensive impression.
  3. What relationship does it have between Prx1+Cells (this section) and integration between the cytoskeleton...(this section)? It is suggested to add one or two sentences in the last of section 2 or in the first of section 3 to articulate.
  4. Again, for the section 4, the authors may need to add a short paragraph to introduce which factors can induce bone formation, and cover all the factors as many as possible, while saying in the review we only discuss a few ones described in the recent development. For the section 5, it covers two types of factors, molecular factors and environmental factors. Again, it is better to add a short paragraph ahead to give a comprehensive background that would be help to attract the readers' interest to read.

Author Response

This review manuscript covers recent development in the molecular mechanisms of bone healing. The authors discussed a few key factor including proteins, genes, influences, and environmental factors related to bone fracture repair. The review was not organized very well and it may need an extensive revision. 

R-1: The authors should add a short paragraph summarizing the categories of bone fracture and possible reasons causing bone to break in the introduction.

AUTHOR: You can see below and in the revised manuscript (IN RED) the categories of bone fracture and possible reasons causing bone to break that have been included in the INTRODUCTION (in line 44, just at the end of line 43).

The categories of bone fracture are the following: Closed or open fractures; complete fractures; displaced fractures; partial fractures; and stress fractures. Besides, some extra terms must be added to describe partial, complete, open and closed fractures. These terms include: avulsion; comminuted; compression; impacted; oblique; spiral; and transverse. Most often, bone fractures happen because the bone runs into a stronger force. Also, repetitive forces – like from running — can fracture a bone (stress fractures). Another reason for fractures is osteoporosis, which weakens bones as you age.

R-1: The authors may add a short paragraph in the introduction to summarize proteins, genes and other factors that they discussed in the later sections, as they did in the conclusion. In this way, it will give readers a comprehensive impression.

AUTHOR: The following paragraph has also been added (IN RED) in the INTRODUCTION (in line 62).

In this article the following topics will be analyzed: The role of a specific population (Prx1+ Cells) and its expression marker (Prx1) in fracture repair; integration between the cytoskeleton and the main molecular pathways in relation to the mechanotransmission mechanisms in osteocytes; factors that induce bone formation (type H vessels, endogenous-exogenous combined bionic periosteum, after BMP2 stimulation preconditioned cells display increased in-vivo bone formation ability, in vivo sequestration of innate small molecules, phagocytic role of macrophages, morin); inducing bone formation with mesenchymal stem cells (leptin, two protein networks, Trb3, Interleukin-1β); factors that delay bone healing (inhibition of estrogen receptor alpha signalling, smoking); studies on diabetes and osteoporosis; an improved method for assessing cell and molecular signals in the reparative callus during fracture healing; a novel experimental model to study fracture healing of the proximal femur; and co-culture systems of osteoblasts and osteoclasts (Table 2).

R-1: What relationship does it have between Prx1+Cells (this section) and integration between the cytoskeleton...(this section)? It is suggested to add one or two sentences in the last of section 2 or in the first of section 3 to articulate.

AUTHOR: At the beginning of section 3 (line 97) we have added some sentences (IN RED).

Bone regeneration happens by two mains ossification processes, endochondral ossification in which the skeletal element first develops as a cartilaginous template that is subsequently replaced by bone, and intramembranous ossification in which mesenchymal cells directly differentiate into bone-forming osteoblasts. The ossification process does not need pre-existing cartilage. Both procedures need the replenishment of the osteogenic or chondrogenic progenitor cells that participate in bone or cartilage formation during normal development and under pathologic conditions, such as fracture healing. In general, osteogenic progenitors distribute in various bone compartments along the bone's outer surface within the periosteum and the inner surface of bone within the endosteum.

R-1: Again, for the section 4, the authors may need to add a short paragraph to introduce which factors can induce bone formation, and cover all the factors as many as possible, while saying in the review we only discuss a few ones described in the recent development. 

For the section 5, it covers two types of factors, molecular factors and environmental factors. Again, it is better to add a short paragraph ahead to give a comprehensive background that would be help to attract the readers' interest to read.

AUTHOR: At the beginning of section 4 (line 115) and section 5 (line 293) we have added short paragraphs (IN RED)

Line 115: Factors that can induce bone formation are the following:  Patient-related factors (age, osteoporosis, smoking, drugs); bone grafting, biochemical stimulation (growth factors, bone morphogenetic proteins, transforming growth factor beta); cell-mediated bone healing (osteogenic stem cells); gene therapy (transfecting bone stem cells with genes encoding for growth factors that stimulate bone healing); and physical factors (electrical stimulation). In this review we only discuss a few ones described in the recent development.

Line 293: Factors that delay bone healing can be divided into local and systemic. The most important local factors are the following: inadequate bone reduction, unstable bone fixation, bone infection, and radiation. The most important systemic factors are the following: patient age (bone healing is faster in children than in adults), nutrition status (sufficient amount of nutrients and vitamins A, B, C and D are essential for the healing of broken bones). Smoking has a negative effect on bone healing. Steroids also can slow down the healing process. Systemic diseases such as hyperthyroidism and renal insufficiency delay fracture healing. Genetic diseases such as Marfan syndrome, Ehler-Danlos syndrome, osteogenesis imperfecta are among the factors affecting bone healing.

Reviewer 2 Report

Rodriguez-Merchan writes an interesting review article where they analyze the new advances that have appeared in the molecular mechanisms of fracture healing. An exhaustive, current review has been carried out, which reveals different aspects, cellular and molecular, that participate in the process. The article is difficult to read due to the abundance of data and the absence of an initial script in the introduction. In this it would be interesting to briefly describe the different elements that are going to be analyzed, which will allow a better follow-up of the reader. Some graphic representation would also be desirable to better understand the concepts expressed.

Author Response

REVIEWER-2

R-2: Rodriguez-Merchan writes an interesting review article where they analyze the new advances that have appeared in the molecular mechanisms of fracture healing. An exhaustive, current review has been carried out, which reveals different aspects, cellular and molecular, that participate in the process. The article is difficult to read due to the abundance of data and the absence of an initial script in the introduction. In this it would be interesting to briefly describe the different elements that are going to be analyzed, which will allow a better follow-up of the reader. Some graphic representation would also be desirable to better understand the concepts expressed.

AUTHOR: The following sentences have been included (line 62) in the INTRODUCTION (IN RED) – This color has been used because Reviewer-1 made the same suggestion.

Besides, a NEW TABLE (now Table 2) summarizing the elements analyzed has been included (see below IN BLUE).

In this article the following topics will be analyzed: The role of a specific population (Prx1+ Cells) and its expression marker (Prx1) in fracture repair; integration between the cytoskeleton and the main molecular pathways in relation to the mechanotransmission mechanisms in osteocytes; factors that induce bone formation (type H vessels, endogenous-exogenous combined bionic periosteum, after BMP2 stimulation preconditioned cells display increased in-vivo bone formation ability, in vivo sequestration of innate small molecules, phagocytic role of macrophages, morin); inducing bone formation with mesenchymal stem cells (leptin, two protein networks, Trb3, Interleukin-1β); factors that delay bone healing (inhibition of estrogen receptor alpha signalling, smoking); studies on diabetes and osteoporosis; an improved method for assessing cell and molecular signals in the reparative callus during fracture healing; a novel experimental model to study fracture healing of the proximal femur; and co-culture systems of osteoblasts and osteoclasts (Table 2).

Table 2. Brief description of the different elements on fracture healing analyzed in this article.

The role of a specific population (Prx1+ Cells) and its expression marker (Prx1)

Integration between the cytoskeleton and the main molecular pathways in relation to the mechanotransmission mechanisms in osteocytes

Factors that induce bone formation

1) Type H vessels

2) Endogenous-exogenous combined bionic periosteum

3) Preconditioned cells (after BMP2 stimulation)

4) In vivo sequestration of innate small molecules  

5) Macrophages (phagocytic role)

6) Morin

7) Mesenchymal stem cells (leptin; two protein networks; Trb3; interleukin-1β)

Factors that delay bone healing

1) Inhibition of estrogen receptor alpha signalling

2)  Smoking

Studies on diabetes and osteoporosis

1) Transcriptomic and proteomic approaches

2) BMP6 deficiency (35 kDa);

3) Skeletal stem cell defect

An improved method for assessing cell and molecular signals in the reparative callus

A novel experimental model to study fracture healing of the proximal femur

Co-culture systems of osteoblasts and osteoclasts

Round 2

Reviewer 1 Report

none